# FEDERATED CONTINUAL LEARNING WITH WEIGHTED INTER-CLIENT TRANSFER

## ABSTRACT

There has been a surge of interest in continual learning and federated learning, both of which are important in deep neural networks in real-world scenarios. Yet little research has been done regarding the scenario where each client learns on a sequence of tasks from a private local data stream. This problem of *federated continual learning* poses new challenges to continual learning, such as utilizing knowledge from other clients, while preventing interference from irrelevant knowledge. To resolve these issues, we propose a novel federated continual learning framework, *Federated Weighted Inter-client Transfer (FedWeIT)*, which decomposes the network weights into global federated parameters and sparse task-specific parameters, and each client receives selective knowledge from other clients by taking a weighted combination of their task-specific parameters. *FedWeIT* minimizes interference between incompatible tasks, and also allows positive knowledge transfer across clients during learning. We validate our *FedWeIT* against existing federated learning and continual learning methods under varying degrees of task similarity across clients, and our model significantly outperforms them with a large reduction in the communication cost.

## 1 INTRODUCTION

Continual learning (Thrun, 1995; Kumar & Daume III, 2012; Ruvolo & Eaton, 2013; Kirkpatrick et al., 2017; Schwarz et al., 2018) describes a learning scenario where a model continuously trains on a sequence of tasks; it is inspired by the human learning process, as a person learns to perform numerous tasks with large diversity over his/her lifespan, making use of the past knowledge to learn about new tasks without forgetting previously learned ones. Continual learning is a long-studied topic since having such an ability leads to the potential of building a general artificial intelligence. However, there are crucial challenges in implementing it with conventional models such as deep neural networks (DNNs), such as *catastrophic forgetting*, which describes the problem where parameters or semantic representations learned for the past tasks drift to the direction of new tasks during training. The problem has been tackled by various prior work (Kirkpatrick et al., 2017; Lee et al., 2017; Shin et al., 2017; Riemer et al., 2019). More recent works tackle other issues, such as scalability or order-robustness (Schwarz et al., 2018; Hung et al., 2019; Yoon et al., 2020).

However, all of these models are fundamentally limited in that the models can only learn from its direct experience - they only learn from the sequence of the tasks they have trained on. Contrarily, humans can learn from *indirect experience* from others, through different means (e.g. verbal communications, books, or various media). Then wouldn't it be beneficial to implement such an ability to a continual learning framework, such that multiple models learning on different machines can learn from the knowledge of the tasks that have been already experienced by other clients? One problem that arises here, is that due to data privacy on individual clients and exorbitant communication cost, it may not be possible to communicate data directly between the clients or between the server and clients. Federated learning (McMahan et al., 2016; Li et al., 2018; Yurochkin et al., 2019) is a learning paradigm that tackles this issue by communicating the parameters instead of the raw data itself. We may have a server that receives the parameters locally trained on multiple clients, aggregates it into a single model parameter, and sends it back to the clients. Motivated by our intuition on learning from indirect experience, we tackle the problem of *Federated Continual Learning (FCL)* where we perform continual learning with multiple clients trained on private task sequences, which communicate their task-specific parameters via a global server. Figure 1 (a) depicts an example

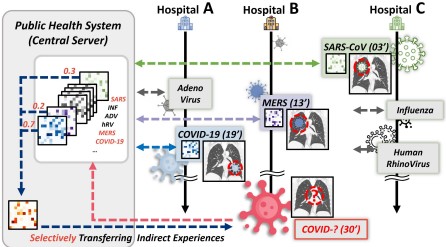 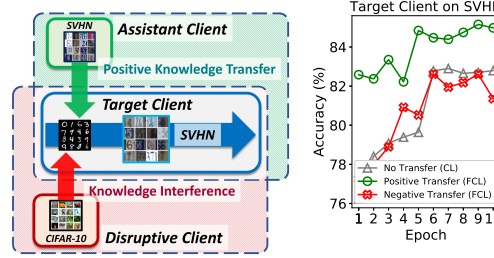

(a) Federated Continual Learning      (b) Challenges of Federated Continual Learning

Figure 1: **(a): Concept**. A continual learner at a hospital which learns on sequence of disease prediction tasks may want to utilize relevant task parameters from other hospitals. FCL allows such inter-client knowledge transfer via the communication of task-decomposed parameters. **(b): Challenge of FCL.** Interference from other clients, resulting from sharing irrelevant knowledge, may hinder an optimal training of target clients (*Red*) while relevant knowledge from other clients will be beneficial for their learning (*Green*).

scenario of FCL. Suppose that we are building a network of hospitals, each of which has a disease diagnosis model which continuously learns to perform diagnosis given CT scans, for new types of diseases. Then, under our framework, any diagnosis model which has learned about a new type of disease (e.g. COVID-19) will transmit the task-specific parameters to the global server, which will redistribute them to other hospitals for the local models to utilize. This allows all participants to benefit from the new task knowledge without compromising the data privacy.

Yet, the problem of federated continual learning also brings new challenges. First, there is not only the catastrophic forgetting from continual learning, but also the **threat of potential interference from other clients**. Figure 1 (b) describes this challenge with the results of a simple experiment. Here, we train a model for MNIST digit recognition while communicating the parameters from another client trained on a different dataset. When the knowledge transferred from the other client is relevant to the target task (SVHN), the model starts with high accuracy, converge faster and reach higher accuracy (**green line**), whereas the model underperforms the base model if the transferred knowledge is from a task highly different from the target task (CIFAR-10, **red line**). Thus, we need to *selective utilize* knowledge from other clients to minimize the *inter-client interference* and maximize *inter-client knowledge transfer*. Another problem with the federated learning is efficient communication, as communication cost could become excessively large when utilizing the knowledge of the other clients, since the communication cost could be the main bottleneck in practical scenarios when working with edge devices. Thus we want the knowledge to be represented as compactly as possible.

To tackle these challenges, we propose a novel framework for federated continual learning, *Federated Weighted Inter-client Transfer (FedWeIT)*, which decomposes the local model parameters into a dense base parameter and sparse task-adaptive parameters. FedWeIT reduces the interference between different tasks since the base parameters will encode task-generic knowledge, while the task-specific knowledge will be encoded into the task-adaptive parameters. When we utilize the generic knowledge, we also want the client to selectively utilize task-specific knowledge obtained at other clients. To this end, we allow each model to take a weighted combination of the task-adaptive parameters broadcast from the server, such that it can select task-specific knowledge helpful for the task at hand. FedWeIT is communication-efficient, since the task-adaptive parameters are *highly sparse* and only need to be communicated once when created. Moreover, when communication efficiency is not a critical issue as in cross-silo federated learning (Kairouz et al., 2019), we can use our framework to incentivize each client based on the attention weights on its task-adaptive parameters. We validate our method on multiple different scenarios with varying degree of task similarity across clients against various federated learning and local continual learning models. The results show that our model obtains significantly superior performance over all baselines, adapts faster to new tasks, with largely reduced communication cost. The main contributions of this paper are as follows:

- We introduce **a new problem of Federated Continual Learning (FCL)**, where multiple models continuously learn on distributed clients, which poses new challenges such as prevention of inter-client interference and inter-client knowledge transfer.

- We propose **a novel and communication-efficient framework for federated continual learning**, which allows each client to adaptively update the federated parameter and selectively utilize the past knowledge from other clients, by communicating sparse parameters.

## 2 RELATED WORK

**Continual learning** While continual learning (Kumar & Daume III, 2012; Ruvolo & Eaton, 2013) is a long-studied topic with a vast literature, we only discuss recent relevant works. **Regularization-based:** EWC (Kirkpatrick et al., 2017) leverages Fisher Information Matrix to restrict the change of the model parameters such that the model finds solution that is good for both previous and the current task, and IMM (Lee et al., 2017) proposes to learn the posterior distribution for multiple tasks as a mixture of Gaussians. **Architecture-based:** DEN (Yoon et al., 2018) tackles this issue by expanding the networks size that are necessary via iterative neuron/filter pruning and splitting, and RCL (Xu & Zhu, 2018) tackles the same problem using reinforcement learning. APD (Yoon et al., 2020) additively decomposes the parameters into shared and task-specific parameters to minimize the increase in the network complexity. **Coreset-based:** GEM variants (Lopez-Paz & Ranzato, 2017; Chaudhry et al., 2019) minimize the loss on both of actual dataset and stored episodic memory. FRCL (Titsias et al., 2020) memorizes approximated posteriors of previous tasks with sophisticatedly constructed inducing points. To the best of our knowledge, none of the existing approaches considered the communicability for continual learning of deep neural networks, which we tackle. CoLLA (Rostami et al., 2018) aims at solving multi-agent lifelong learning with sparse dictionary learning, but it is not applicable to federated learning or continual deep learning.

**Federated learning** Federated learning is a distributed learning framework under differential privacy, which aims to learn a global model on a server while aggregating the parameters learned at the clients on their private data. FedAvg (McMahan et al., 2016) aggregates the model trained across multiple clients by computing a weighted average of them based on the number of data points trained. FedProx (Li et al., 2018) trains the local models with a proximal term which restricts their updates to be close to the global model. FedCurv (Shoham et al., 2019) aims to minimize the model disparity across clients during federated learning by adopting a modified version of EWC. Recent works Yurochkin et al. (2019); Wang et al. (2020) introduce well-designed aggregation policies by leveraging Bayesian non-parametric methods. A crucial challenge of federated learning is **the reduction of communication cost**. TWAFL (Chen et al., 2019) tackles this problem by performing layer-wise parameter aggregation, where shallow layers are aggregated at every step, but deep layers are aggregated in the last few steps of a loop. Karimireddy et al. (2020) suggests an algorithm for rapid convergence, which minimizes the interference among discrepant tasks at clients by sacrificing the local optimality. This is an opposite direction from personalized federated learning methods (Fallah et al., 2020; Lange et al., 2020; Deng et al., 2020) which put more emphasis on the performance of local models. FCL is a parallel research direction to both, and to the best of our knowledge, ours is the first work that considers task-incremental learning of clients under federated learning framework.

## 3 FEDERATED CONTINUAL LEARNING WITH FEDWEIT

Motivated by the human learning process from indirect experiences, we introduce a novel continual learning under federated learning setting, which we refer to as *Federated Continual Learning (FCL)*. FCL assumes that multiple clients are trained on a sequence of tasks from private data stream, while communicating the learned parameters with a global server. We first formally define the problem in *Section* 3.1, and then propose naive solutions that straightforwardly combine the existing federated learning and continual learning methods in *Section* 3.2. Then, following *Section* 3.3 and 3.4, we discuss about two novel challenges that are introduced by federated continual learning, and propose a novel framework, *Federated Weighted Inter-client Transfer (FedWeIT)* which can effectively handle the two problems while also reducing the client-to-sever communication cost.

### 3.1 PROBLEM DEFINITION

In the standard continual learning (on a single machine), the model iteratively learns from a sequence of tasks $\{\mathcal{T}^{(1)}, \mathcal{T}^{(2)}, ..., \mathcal{T}^{(T)}\}$ where $\mathcal{T}^{(t)}$ is a labeled dataset of $t^{th}$ task, $\mathcal{T}^{(t)} = \{\mathbf{x}_i^{(t)}, \mathbf{y}_i^{(t)}\}_{i=1}^{N_t}$, which consists of $N_t$ pairs of instances $\mathbf{x}_i^{(t)}$ and their corresponding labels $\mathbf{y}_i^{(t)}$. Assuming the most realistic situation, we consider the case where the task sequence is a task stream with an unknown arriving order, such that the model can access $\mathcal{T}^{(t)}$ only at the training period of task $t$ which becomes inaccessible afterwards. Given $\mathcal{T}^{(t)}$ and the model learned so far, the learning objective at task $t$ is as follows: $\text{minimize}_{\boldsymbol{\theta}^{(t)}} \mathcal{L}(\boldsymbol{\theta}^{(t)}; \boldsymbol{\theta}^{(t-1)}, \mathcal{T}^{(t)})$, where $\boldsymbol{\theta}^{(t)}$ is a set of the model parameters at task $t$.

We now extend the conventional continual learning to the federated learning setting with multiple clients and a global server. Let us assume that we have $C$ clients, where at each client $c_c \in \{c_1, \ldots, c_C\}$ trains a model on a *privately accessible* sequence of tasks $\{\mathcal{T}_c^{(1)}, \mathcal{T}_c^{(2)}, ..., \mathcal{T}_c^{(t)}\} \subseteq \mathcal{T}$. Please note that there is no relation among the tasks $\mathcal{T}_{1:c}^{(t)}$ received at step $t$, across clients. Now the goal is to effectively train $C$ continual learning models on their own private task streams, via communicating the model parameters with the global server, which aggregates the parameters sent from each client, and redistributes them to clients.

## 3.2 COMMUNICABLE CONTINUAL LEARNING

In conventional federated learning settings, the learning is done with multiple rounds of local learning and parameter aggregation. At each round of communication $r$, each client $c_c$ and the server $s$ perform the following two procedures: *local parameter transmission* and *parameter aggregation & broadcasting*. In the local parameter transmission step, for a randomly selected subset of clients at round $r$, $\mathcal{C}^{(r)} \subseteq \{c_1, c_2, ..., c_C\}$, each client $c_c$ sends updated parameters $\boldsymbol{\theta}^{(r)}$ to the server. The server-clients transmission is not done at every client because some of the clients may be temporarily disconnected. Then the server aggregates the parameters $\boldsymbol{\theta}_c^{(r)}$ sent from the clients into a single parameter. The most popular frameworks for this aggregation are FedAvg (McMahan et al., 2016) and FedProx (Li et al., 2018). However, naive federated continual learning with these two algorithms on local sequences of tasks may result in catastrophic forgetting. One simple solution is to use a regularization-based, such as Elastic Weight Consolidation (EWC) (Kirkpatrick et al., 2017), which allows the model to obtain a solution that is optimal for both the previous and the current tasks. There exist other advanced solutions (Rusu et al., 2016; Nguyen et al., 2018; Chaudhry et al., 2019) that successfully prevents catastrophic forgetting. However, the prevention of catastrophic forgetting at the client level is an orthogonal problem from federated learning.

Thus we focus on challenges that newly arise in this federated continual learning setting. In the federated continual learning framework, the aggregation of the parameters into a global parameter $\boldsymbol{\theta}_G$ allows inter-client knowledge transfer across clients, since a task $\mathcal{T}_i^{(q)}$ learned at client $c_i$ at round $q$ may be similar or related to $\mathcal{T}_j^{(r)}$ learned at client $c_j$ at round $r$. Yet, using a single aggregated parameter $\boldsymbol{\theta}_G$ may be suboptimal in achieving this goal since knowledge from irrelevant tasks may not to be useful or even hinder the training at each client by altering its parameters into incorrect directions, which we describe as *inter-client interference*. Another problem that is also practically important, is the *communication-efficiency*. Both the parameter transmission from the client to the server, and server to client will incur large communication cost, which will be problematic for the continual learning setting, since the clients may train on possibly unlimited streams of tasks.

## 3.3 FEDERATED WEIGHTED INTER-CLIENT TRANSFER

How can we then maximize the *knowledge transfer* between clients while minimizing the *inter-client interference*, and communication cost? We now describe our model, *Federated Weighted Inter-client Transfer (FedWeIT)*, which can resolve the these two problems that arise with a naive combination of continual learning approaches with federated learning framework.

The main cause of the problems, as briefly alluded to earlier, is that the knowledge of all tasks learned at multiple clients is stored into a single set of parameters $\boldsymbol{\theta}_G$. However, for the knowledge transfer to be effective, each client should *selectively* utilize only the knowledge of the *relevant* tasks that is trained at other clients. This **selective transfer** is also the key to minimize the inter-client interference as well as it will disregard the knowledge of irrelevant tasks that may interfere with learning.

We tackle this problem by decomposing the parameters, into three different types of the parameters with different roles: *global parameters* ($\boldsymbol{\theta}_G$) that capture the global and generic knowledge across all clients, *local base parameters* (**B**) which capture generic knowledge for each client, and *task-adaptive parameters* (**A**) for each specific task per client, motivated by Yoon et al. (2020). A set of the model parameters $\boldsymbol{\theta}_c^{(t)}$ for task $t$ at continual learning client $c_c$ is then defined as follows:

$$\boldsymbol{\theta}_c^{(t)} = \mathbf{B}_c^{(t)} \odot \mathbf{m}_c^{(t)} + \mathbf{A}_c^{(t)} + \sum_{i \in \mathcal{C}_{\setminus c}} \sum_{j < |t|} \alpha_{i,j}^{(t)} \mathbf{A}_i^{(j)} \tag{1}$$

where $\mathbf{B}_c^{(t)} \in \{\mathbb{R}^{I_l \times O_l}\}_{l=1}^L$ is the set of base parameters for $c^{th}$ client shared across all tasks in the

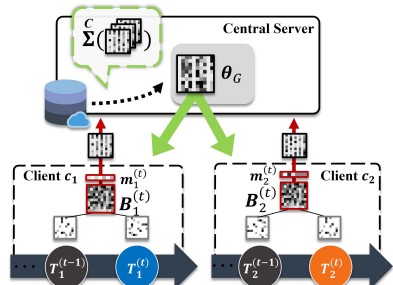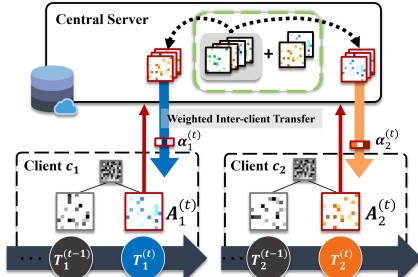

(a) Communication of General Knowledge  (b) Communication of Task-adaptive Knowledge

Figure 2: Updates of *FedWeIT*. **(a)** A client sends sparsified federated parameter $\mathbf{B}_c \odot \mathbf{m}_c^{(t)}$. After that, the server redistributes aggregated parameters to the clients. **(b)** The knowledge base stores previous tasks-adaptive parameters of clients, and each client selectively utilizes them with an attention mask.

client, $\mathbf{m}_c^{(t)} \in \{\mathbb{R}^{O_l}\}_{l=1}^{L}$ is the set of sparse vector masks which allows to adaptively transform $\mathbf{B}_c^{(t)}$ for the task $t$, $\mathbf{A}_c^{(t)} \in \{\mathbb{R}^{I_l \times O_l}\}_{l=1}^{L}$ is the set of a sparse task-adaptive parameters at client $c_c$. Here, $L$ is the number of the layer in the neural network, and $I_l, O_l$ are input and output dimension of the weights at layer $l$, respectively.

The first term allows selective utilization of the global knowledge. We want the base parameter $\mathbf{B}_c^{(t)}$ at each client to capture generic knowledge across all tasks across all clients. In Figure 2 (a), we initialize it at each round $t$ with the global parameter from the previous iteration, $\boldsymbol{\theta}_G^{(t-1)}$ which aggregates the parameters sent from the client. This allows $\mathbf{B}_c^{(t)}$ to also benefit from the *global* knowledge about all the tasks. However, since $\boldsymbol{\theta}_G^{(t-1)}$ also contains knowledge irrelevant to the current task, instead of using it as is, we learn the sparse mask $\mathbf{m}_c^{(t)}$ to select only the relevant parameters for the given task. This sparse parameter selection helps minimize inter-client interference, and also allows for efficient communication. The second term is the task-adaptive parameters $\mathbf{A}_c^{(t)}$. Since we additively decompose the parameters, this will learn to capture knowledge about the task that is not captured by the first term, and thus will capture specific knowledge about the task $\mathcal{T}_c^{(t)}$. The final term describes weighted inter-client knowledge transfer. We have a set of parameters that are *transmitted* from the server, which contain all task-adaptive parameters from all the clients. To selectively utilizes these indirect experiences from other clients, we further allocate attention $\boldsymbol{\alpha}_c^{(t)}$ on these parameters, to take a weighted combination of them. By learning this attention, each client can select only the relevant task-adaptive parameters that help learn the given task. Although we design $A_i^{(j)}$ to be highly sparse, using about $2-3\%$ of memory of full parameter in practice, sending all task knowledge is not desirable. Thus we only transmit the task-adaptive parameter of the previous task $(t-1)$, which we empirically find to achieve good results in practice.

**Training.**  We learn the decomposable parameter $\boldsymbol{\theta}_c^{(t)}$ by optimizing for the following objective:

$$\underset{\mathbf{B}_c^{(t)},\; \mathbf{m}_c^{(t)},\; \mathbf{A}_c^{(1:t)},\; \boldsymbol{\alpha}_c^{(t)}}{\text{minimize}} \quad \mathcal{L}\left(\boldsymbol{\theta}_c^{(t)}; \mathcal{T}_c^{(t)}\right) + \lambda_1 \Omega(\{\mathbf{m}_c^{(t)}, \mathbf{A}_c^{(1:t)}\}) + \lambda_2 \sum_{i=1}^{t-1} \|\Delta\mathbf{B}_c^{(t)} \odot \mathbf{m}_c^{(i)} + \Delta\mathbf{A}_c^{(i)}\|_2^2, \quad (2)$$

where $\mathcal{L}$ is a loss function and $\Omega(\cdot)$ is a sparsity-inducing regularization term for all task-adaptive parameters and the masking variable (we use $\ell_1$-norm regularization), to make them sparse. The second regularization term is used for retroactive update of the past task-adaptive parameters, which helps the task-adaptive parameters to maintain the original solutions for the target tasks, by reflecting the change of the base parameter. Here, $\Delta\mathbf{B}_c^{(t)} = \mathbf{B}_c^{(t)} - \mathbf{B}_c^{(t-1)}$ is the difference between the base parameter at the current and previous timestep, and $\Delta\mathbf{A}_c^{(i)}$ is the difference between the task-adaptive parameter for task $i$ at the current and previous timestep. This regularization is essential for preventing catastrophic forgetting. $\lambda_1$ and $\lambda_2$ are hyperparameters controlling the effect of the two regularizers.

### 3.4 EFFICIENT COMMUNICATION VIA SPARSE PARAMETERS

FedWeIT learns via server-to-client communication. As discussed earlier, a crucial challenge here is to reduce the communication cost. We describe what happens at the client and the server at each step.

**Algorithm 1** Federated Weighted Inter-client Transfer

**input** Dataset $\{\mathcal{D}_c^{(1:t)}\}_{c=1}^C$, and Global Parameter $\boldsymbol{\theta}_G$
**output** $\{\mathbf{B}_c, \mathbf{m}_c^{(1:t)}, \boldsymbol{\alpha}_c^{(1:t)}, \mathbf{A}_c^{(1:t)}\}_{c=1}^C$
1: Initialize $\mathbf{B}_c$ to $\boldsymbol{\theta}_G$ for all $c \in \mathcal{C} \equiv \{1, ..., C\}$
2: **for** task $t = 1, 2, ...$ **do**
3:     **for** round $r = 1, 2, ..., R$ **do**
4:         Transmit $\widehat{\mathbf{B}}_c^{(t,r)}$ and $\mathbf{A}_c^{(t-1,R)}$ of client $c_c$ to server
5:         Compute $\boldsymbol{\theta}_G^{(r)} \leftarrow \frac{1}{|\mathcal{C}|} \sum_{c \in \mathcal{C}} \widehat{\mathbf{B}}_c^{(t,r)}$
6:         Distribute $\boldsymbol{\theta}_G^{(r)}$ and $\{\mathbf{A}_j^{(t-1,R)}\}_{j \in \mathcal{C}}$ to client $c$
7:         Minimize Eq. (2) for solving each local CL problems
8:     **end for**
9: **end for**

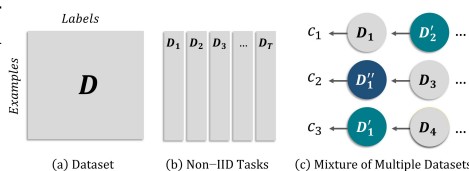

Figure 3: **Configuration of task sequences:** We first split a dataset $D$ into multiple sub-tasks in non-IID manner ((a) and (b)). Then, we distribute them to multiple clients ($C_\#$). Mixed tasks from multiple datasets (colored circles) are distributed across all clients ((c)).

**Client:** At each round $r$, each client $c_c$ partially updates its base parameter with the nonzero components of the global parameter sent from the server; that is, $\mathbf{B}_c(n) = \boldsymbol{\theta}_G(n)$ where $n$ is a nonzero element of the global parameter. After training the model using Eq. (2), it obtains a sparsified base parameter $\widehat{\mathbf{B}}_c^{(t)} = \mathbf{B}_c^{(t)} \odot \mathbf{m}_c^{(t)}$ and task-adaptive parameter $\mathbf{A}_c^{(t)}$ for the new task, both of which are sent to the server, at smaller cost compared to naive FCL baselines. While naive FCL baselines require $|\mathcal{C}| \times R \times |\boldsymbol{\theta}|$ for client-to-server communication, FedWeIT requires $|\mathcal{C}| \times (R \times |\widehat{\mathbf{B}}| + |\mathbf{A}|)$ where $R$ is the number of communication round per task and $|\cdot|$ is the number of parameters.

**Server:** The server first aggregates the base parameters sent from all the clients by taking an weighted average of them: $\boldsymbol{\theta}_G = \frac{1}{C} \sum_{\mathcal{C}} \widehat{\mathbf{B}}_i^{(t)}$. Then, it broadcasts $\boldsymbol{\theta}_G$ to all the clients. Task adaptive parameters of $t - 1$, $\{\mathbf{A}_i^{(t-1)}\}_{i=1}^{\mathcal{C}_{\backslash c}}$ are broadcast at once per client during training task $t$. While naive FCL baselines requires $|\mathcal{C}| \times R \times |\boldsymbol{\theta}|$ for server-to-client communication cost, FedWeIT requires $|\mathcal{C}| \times (R \times |\boldsymbol{\theta}_G| + (|\mathcal{C}| - 1) \times |\mathbf{A}|)$ in which $\boldsymbol{\theta}_G, \mathbf{A}$ are highly sparse. We describe the FedWeIT algorithm in Algorithm 1. For a detailed version of the algorithm, please see Section D in appendix.

## 4 EXPERIMENTS

We validate our **FedWeIT** under different configurations of task sequences against baselines which are namely Overlapped-CIFAR-100 and NonIID-50. **1) Overlapped-CIFAR-100**: We group 100 classes of CIFAR-100 dataset into 20 non-iid superclasses tasks. Then, we randomly sample 10 tasks out of 20 tasks and split instances to create a task sequence for each of the clients with overlapping tasks. **2) NonIID-50**: We use the following eight benchmark datasets: MNIST (LeCun et al., 1998), CIFAR-10/-100 (Krizhevsky & Hinton, 2009), SVHN (Netzer et al., 2011), Fashion-MNIST (Xiao et al., 2017), Not-MNIST (Bulatov, 2011), FaceScrub (Ng & Winkler, 2014), and TrafficSigns (Stallkamp et al., 2011). We split the classes in the 8 datasets into 50 non-IID tasks, each of which is composed of 5 classes that are disjoint from the classes used for the other tasks. This is a large-scale experiment, containing $280,000$ images of 293 classes from 8 heterogeneous datasets. After generating and processing tasks, we randomly distribute them to multiple clients as illustrated in Figure 3.

**Experimental setup** We use a modified version of LeNet (LeCun et al., 1998) for the experiments with both Overlapped-CIFAR-100 and NonIID-50 dataset. Further, we use ResNet-18 He et al. (2016) with NonIID-50 dataset. We followed other experimental setups from Serrà et al. (2018) and Yoon et al. (2020). For detailed descriptions of the task configuration and hyperparameters used, please see Section B in appendix. Also, for more ablation studies, please see Section C in appendix.

**Baselines and our model** **1) STL**: Single Task Learning at each arriving task. **2) Local-EWC**: Individual continual learning with *EWC* (Kirkpatrick et al., 2017) per client. **3) Local-APD**: Individual continual learning with *APD* (Yoon et al., 2020) per client. **4) FedProx**: FCL using *FedProx* (Li et al., 2018) algorithm. **5) Scaffold**: FCL using *Scaffold* (Karimireddy et al., 2020) algorithm. **6) FedCurv**: FCL using *FedCurv* (Shoham et al., 2019) algorithm. **7) FedProx-[model]**: FCL, that is trained using *FedProx* algorithm with [model]. **8) FedWeIT**: Our FedWeIT algorithm.

Table 1: Averaged Per-task performance on both dataset during FCL with 5 clients (fraction=1.0). We measured task accuracy and model size after completing all learning phases over 3 individual trials. We also measured C2S/S2C communication cost for training each task.

| Methods | NonIID-50 Dataset ($F$=1.0, $R$=20) | | | Overlapped CIFAR-100 ($F$=1.0, $R$=20) | | |
|---|---|---|---|---|---|---|
| | Accuracy | Model Size | C2S/S2C Cost | Accuracy | Model Size | C2S/S2C Cost |
| STL | $85.78 \pm 0.17$ | 0.610 GB | N/A | $57.15 \pm 0.07$ | 0.610 GB | N/A |
| Local-EWC | $74.30 \pm 0.08$ | 0.061 GB | N/A | $44.26 \pm 0.43$ | 0.061 GB | N/A |
| Local-APD | $81.42 \pm 0.72$ | 0.090 GB | N/A | $50.82 \pm 0.33$ | 0.073 GB | N/A |
| FedProx | $63.69 \pm 1.75$ | 0.061 GB | 1.22 / 1.22 GB | $33.83 \pm 0.48$ | 0.061 GB | 1.22 / 1.22 GB |
| Scaffold | $30.84 \pm 1.41$ | 0.061 GB | 2.44 / 2.44 GB | $22.80 \pm 0.47$ | 0.061 GB | 2.44 / 2.44 GB |
| FedCurv | $72.39 \pm 0.32$ | 0.061 GB | 1.22 / 1.22 GB | $40.36 \pm 0.44$ | 0.061 GB | 1.22 / 1.22 GB |
| FedProx-EWC | $68.18 \pm 0.58$ | 0.061 GB | 1.22 / 1.22 GB | $41.91 \pm 0.47$ | 0.061 GB | 1.22 / 1.22 GB |
| FedProx-APD | $81.20 \pm 1.24$ | 0.079 GB | 1.22 / 1.22 GB | $52.20 \pm 0.41$ | 0.075 GB | 1.22 / 1.22 GB |
| FedWeIT | $\mathbf{84.11} \pm \mathbf{0.27}$ | 0.078 GB | **0.37 / 1.07** GB | $\mathbf{55.16} \pm \mathbf{0.19}$ | 0.075 GB | **0.37 / 1.07** GB |

| 100 clients ($F$=0.05, $R$=20, **1,000 tasks** in total) | | | |
|---|---|---|---|
| Methods | Accuracy | Model Size | C2S/S2C Cost |
| STL | $32.96 \pm 0.23$ | 12.20 GB | N/A |
| Local-APD | $37.50 \pm 0.17$ | 4.01 GB | N/A |
| FedProx | $24.11 \pm 0.44$ | 1.22 GB | 1.22 / 1.22 GB |
| FedCurv | $29.11 \pm 0.20$ | 1.22 GB | 1.22 / 1.22 GB |
| FedCurv-EWC | $29.72 \pm 0.20$ | 1.22 GB | 1.22 / 1.22 GB |
| FedWeIT | $\mathbf{39.58} \pm \mathbf{0.27}$ | 4.03 GB | 0.38 / 1.10 GB |

Figure 4: **Left:** Averaged task adaptation during training last two ($9^{th}$ and $10^{th}$) tasks with 5 and 100 clients. **Right:** Average Per-task Performance on Overlapped-CIFAR-100 during FCL with 100 clients.

## 4.1 EXPERIMENTAL RESULTS

We first validate our model on both Overlapped-CIFAR-100 and NonIID-50 task sequences against single task learning (STL), continual learning (EWC, APD), federated learning (FedProx, Scaffold, FedCurv), and naive federated continual learning (FedProx-based) baselines. Table 1 shows the final average per-task performance after the completion of (federated) continual learning on both datasets. We observe that FedProx-based federated continual learning (FCL) approaches degenerate the performance of continual learning (CL) methods over the same methods without federated learning. This is because the aggregation of all client parameters that are learned on irrelevant tasks results in severe interference in the learning for each task, which leads to catastrophic forgetting and suboptimal task adaptation. Scaffold achieves poor performance on FCL, as its regularization on the local gradients is harmful for FCL, where all clients learn from a different task sequences. While FedCurv reduces inter-task disparity in parameters, it cannot minimize inter-task interference, which results it to underperform single-machine CL methods. On the other hand, FedWeIT significantly outperforms both single-machine CL baselines and naive FCL baselines on both datasets. Even with larger number of clients ($C = 100$), FedWeIT consistently outperforms all baselines (Figure 4). This improvement largely owes to FedWeIT's ability to selectively utilize the knowledge from other clients to rapidly adapt to the target task, and obtain better final performance (Figure 4 Left).

The fast adaptation to new task is another clear advantage of inter-client knowledge transfer. To further demonstrate the practicality of our method with larger networks, we experiment on Non-IID dtaset with ResNet-18 (Table 2), on which FedWeIT still significantly outperforms the strongest baseline (FedProx-APD) while using fewer parameters. Also, our model is not sensitive to the hyperparameters $\lambda_1$ and $\lambda_2$, if they are within reasonable scales (Figure 6 Left).

Table 2: FCL results on NonIID-50 dataset with ResNet-18.

| Methods | ResNet-18 | |
|---|---|---|
| | Acc. | M Size |
| Local-APD | 92.44 % | 1.86 GB |
| FedProx-APD | 92.89 % | 2.05 GB |
| FedWeIT | **94.86 %** | **1.84** GB |

**Efficiency of FedWeIT** We also report the accuracy as a function of network capacity in Table 1, 2, which we measure by the number of parameters used. We observe that FedWeIT obtains much higher accuracy while utilizing less number of parameters compared to FedProx-APD. This efficiency mainly comes from the reuse of task-adaptive parameters from other clients, which is not possible with single-machine CL methods or naive FCL methods. We also examine the communication cost (the size of non-zero parameters transmitted) of each method. Table 1 reports both the *client-to-server* (C2S) / *server-to-client* (S2C) communication cost at training each task. FedWeIT, uses only 30%

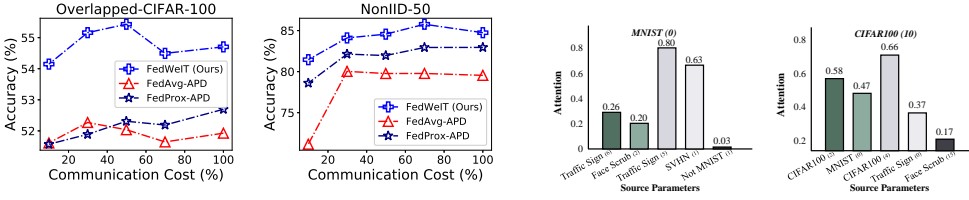

(a) Accuracy over client-to-server cost    (b) Inter-client Knowledge Transfer of *FedWeIT*

Figure 5: **(a) Accuracy over C2S cost.** We report the relative communication cost to the original network. All results are averaged over the 5 clients. **(b) Inter-client transfer** for NonIID-50. We compare the scale of the attentions at first FC layer which gives the weights on transferred task-adaptive parameters from other clients.

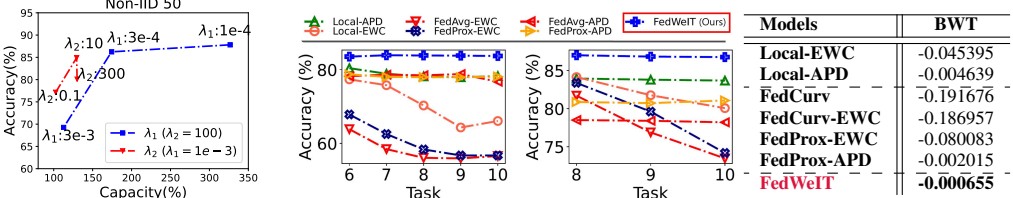

Figure 6: **Left:** Performance of FedWeIT with different scale of hyperparameters on *Non-iid 50*. **Middle:** Performance comparison about current task adaptation at $6^{th}$ and $8^{th}$ tasks during federated continual learning on *NonIID-50*. **Right:** Forgetting measure using Backward Transfer (BWT).

and $3\%$ of parameters for $\widehat{\mathbf{B}}$ and $\mathbf{A}$ of the dense models respectively. We observe that FedWeIT is significantly more communication-efficient than FCL baselines although it broadcasts task-adaptive parameters, due to high sparsity of the parameters. Figure 5 (a) shows the accuracy as a function of C2S cost according to a transmission of top-$\kappa\%$ informative parameters. Since FedWeIT selectively utilizes task-specific parameters learned from other clients, it results in superior performance over APD-baselines especially with sparse communication of model parameters.

**Catastrophic forgetting**  Further, we examine how the performance of the past tasks change during continual learning, to see the severity of catastrophic forgetting with each method. Figure 6 Left shows the performance of FedWeIT and FCL baselines on the $6^{th}$ and $8^{th}$ tasks, at the end of training for later tasks. We observe that naive FCL baselines suffer from more severe catastrophic forgetting than local continual learning with EWC because of the *inter-client interference*, where the knowledge of irrelevant tasks from other clients overwrites the knowledge of the past tasks. Contrarily, our model shows no sign of catastrophic forgetting. This is mainly due to the selective utilization of the prior knowledge learned from other clients through the global/task-adaptive parameters, which allows it to effectively alleviate *inter-client interference*. FedProx-APD also does not suffer from catastrophic forgetting, but they yield inferior performance due to ineffective knowledge transfer. We also report *Backward Transfer (BWT)*, which is a measure on catastrophic forgetting for all models (more positive the better). We provide the details of BWT in the Section B in appendix.

**Weighted inter-client knowledge transfer**  By analyzing the attention $\boldsymbol{\alpha}$ in Eq. (1), we examine which task parameters from other clients each client selected. Figure 5 (b), shows example of the attention weights that are learned for the $0^{th}$ split of *MNIST* and $10^{th}$ split of *CIFAR-100*. We observe that large attentions are allocated to the task parameters from the same dataset (CIFAR-100 utilizes parameters from CIFAR-100 tasks with disjoint classes), or from a similar dataset (MNIST utilizes parameters from Traffic Sign and SVHN). This shows that FedWeIT effectively selects beneficial parameters to maximize *inter-client* knowledge transfer. This is an impressive result since it does not know which datasets the parameters are trained on.

## 5 CONCLUSION

We tackled a novel problem of federated continual learning, whose goal is to continuously learn local models at each client while allowing it to utilize indirect experience (task knowledge) from other clients. This poses new challenges such as *inter-client knowledge transfer* and prevention of *inter-client interference* between irrelevant tasks. To tackle these challenges, we additionally decomposed the model parameters at each client into the global parameters that are shared across all clients, and sparse local task-adaptive parameters that are specific to each task. Further, we allowed each

model to selectively update the global task-shared parameters and selectively utilize the task-adaptive parameters from other clients. The experimental validation of our model under various task similarity across clients, against existing federated learning and continual learning baselines shows that our model obtains significantly outperforms baselines with reduced communication cost. We believe that federated continual learning is a practically important topic of large interests to both research communities of continual learning and federated learning, that will lead to new research directions.

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

# A  APPENDIX

**Organization**    The appendix is organized as follows: In Section B, we further describe the experimental details, including the network architecture, hyper-parameter configurations, forgetting measures, and datasets. Also, we report additional experimental results in Section C about the effect of the communication frequency (Section C.1) and an additional ablation studies for model components on Overlapped-CIFAR-100 dataset (Section C.2). We include a detailed algorithm for our FedWeIT in Section D.

# B  EXPERIMENTAL DETAILS

We further provide the experimental settings in detail, including the descriptions of the network architectures, hyperparameters, and dataset configuration.

**Network architecture**    We utilize a modified version of LeNet and a conventional ResNet-18 as the backbone network architectures for validation. In the LeNet, the first two layers are convolutional neural layers of 20 and 50 filters with the $5 \times 5$ convolutional kernels, which are followed by the two fully-connected layers of 800 and 500 units each. Rectified linear units activations and local response normalization are subsequently applied to each layers. We use $2 \times 2$ max-pooling after each convolutional layer. All layers are initialized based on the variance scaling method. Detailed description of the architecture for LeNet is given in Table 3.

**Configurations**    We use an Adam optimizer with adaptive learning rate decay, which decays the learning rate by a factor of 3 for every 5 epochs with no consecutive decrease in the validation loss. We stop training in advance and start learning the next task (if available) when the learning rate reaches $\rho$. The experiment for LeNet with 5 clients, we initialize by $1e^{-3} \times \frac{1}{3}$ at the beginning of each new task and $\rho = 1e^{-7}$. Mini-batch size is 100, the rounds per task is 20, an the epoch per round is 1. The setting for ResNet-18 is identical, excluding the initial learning rate, $1e^{-4}$. In the case of

Table 3: Base Network Architecture (LeNet) and Total Number of Parameters of both FedWeIT and All Baseline Models. $T$ describes the number of arrived tasks in continual learning.

| Layer | Filter Shape | Stride | Output |
|-------|-------------|--------|--------|
| Input | N/A | N/A | $32 \times 32 \times 3$ |
| Conv 1 | $5 \times 5 \times 20$ | 1 | $32 \times 32 \times 20$ |
| Max Pooling 1 | $3 \times 3$ | 2 | $16 \times 16 \times 20$ |
| Conv 2 | $5 \times 5 \times 50$ | 1 | $16 \times 16 \times 50$ |
| Max Pooling 2 | $3 \times 3$ | 2 | $8 \times 8 \times 50$ |
| Flatten | 3200 | N/A | $1 \times 1 \times 3200$ |
| FC 1 | 800 | N/A | $1 \times 1 \times 800$ |
| FC 2 | 500 | N/A | $1 \times 1 \times 500$ |
| Softmax | Classifier | N/A | $1 \times 1 \times 5 \times T$ |
| **Total Number of Parameters** | | | 3,012,920 |

experiments with 20 and 100 clients, we set the same settings except reducing minibatch size from 100 to 10 with an initial learning rate $1e^{-4}$. We use client fraction 0.25 and 0.05, respectively, at each communication round. we set $\lambda_1 = [1e^{-1}, 4e^{-1}]$ and $\lambda_2 = 100$ for all experiments. Further, we use $\mu = 5e^{-3}$ for FedProx, $\lambda = [1e^{-2}, 1.0]$ for EWC and FedCurv. We initialize the attention parameter $\boldsymbol{\alpha}_c^{(t)}$ as sum to one, $\alpha_{c,j}^{(t)} \leftarrow 1/|\boldsymbol{\alpha}_c^{(t)}|$.

**Backward-transfer (BWT)**    Backward transfer (BWT) is a measure for catastrophic forgetting. BWT compares the performance disparity of previous tasks after learning current task as follows:

$$\text{BWT} = \frac{1}{T-1} \sum_{i<T} P_i^{(T)} - P_i^{(i)}, \tag{3}$$

where $P_i^{(T)}$ is the performance of task $i$ after task $T$ is learned ($i < T$). Thus, a large negative backward transfer value indicates that the performance has been substantially reduced, in which case catastrophic forgetting has happened.

For NonIID-50 dataset, we utilize 8 heterogenous datasets and create 50 non-iid tasks in total as shown in Table 4. Then we arbitrarily select 10 tasks without duplication and distribtue them to 5 clients. The average performance of single task learning on the dataset is $85.78 \pm 0.17(\%)$, measured by our base LeNet architecture.

Table 4: Detailed configuration of *NonIID-50* Dataset.

| | | | | | | |
|---|---|---|---|---|---|---|
| **NonIID-50** | | | | | | |
| **Dataset** | **# Classes** | **# Tasks** | **# Classes (Task)** | **# Train Set** | **# Valid Set** | **# Test Set** |
| **CIFAR-100** | 100 | 15 | 5 | 36,750 | 10,500 | 5,250 |
| **Face Scrub** | 100 | 16 | 5 | 13,859 | 3,959 | 1,979 |
| **Traffic Signs** | 43 | 9 | 5 (3) | 32,170 | 9,191 | 4,595 |
| **SVHN** | 10 | 2 | 5 | 61,810 | 17,660 | 8,830 |
| **MNIST** | 10 | 2 | 5 | 42,700 | 12,200 | 6,100 |
| **CIFAR-10** | 10 | 2 | 5 | 36,750 | 10,500 | 5,250 |
| **Not MNIST** | 10 | 2 | 5 | 11,339 | 3,239 | 1,619 |
| **Fashion MNIST** | 10 | 2 | 5 | 42,700 | 12,200 | 6,100 |
| **Total** | 293 | 50 | 248 | 278,078 | 39,723 | 79,449 |

**Datasets**   We create both Overlapped-CIFAR-100 and NonIID-50 datasets. For Overlapped-CIFAR-100, we generate 20 non-iid tasks based on 20 superclasses, which hold 5 subclasses. We split instances of 20 tasks according to the number of clients (5, 20, and 100) and then distribute the tasks across all clients.

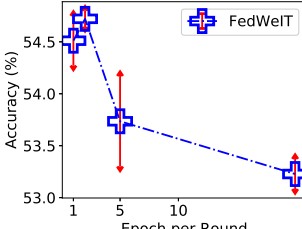

| | Overlapped-CIFAR-100 | | | |
|---|---|---|---|---|
| **Methods** | **Accuracy** | **Model Size** | **C2S/S2C Cost** | **Epochs / Round** |
| **FedWeIT** | **55.16** ± 0.19 | **0.075** GB | 0.37 / 1.07 GB | **1** |
| **FedWeIT** | **55.18** ± 0.08 | 0.077 GB | 0.19 / 0.53 GB | **2** |
| **FedWeIT** | 53.73 ± 0.44 | 0.083 GB | 0.08 / 0.22 GB | **5** |
| **FedWeIT** | 53.22 ± 0.14 | 0.088 GB | 0.02 / 0.07 GB | **20** |

Figure 7: **Average Per-task Performance with error bars over the number of training epochs per communication rounds** on *Overlapped-CIFAR-100* for FedWeIT with 5 clients. All models transmit full of local base parameters and highly sparse task-adaptive parameters. All results are the mean accuracy over 5 clients and we run 3 individual trials. Red arrows at each point describes the standard deviation of the performance.

## C   ADDITIONAL EXPERIMENTAL RESULTS

We further include a quantitative analysis about the communication round frequency and additional experimental results across the number of clients.

### C.1   EFFECT OF THE COMMUNICATION FREQUENCY

We provide an analysis on the effect of the communication frequency by comparing the performance of the model, measured by the number of training epochs per communication round. We run the 4 different FedWeIT with 1, 2, 5, and 20 training epochs per round. Figure 7 shows the performance of our FedWeIT variants. As clients frequently update the model parameters through the communication with the central server, the model gets higher performance while maintaining smaller network capacity since the model with a frequent communication efficiently updates the model parameters as transferring the inter-client knowledge. However, it requires much heavier communication costs than the model with sparser communication. For example, the model trained for 1 epochs at each round may need to about 16.9 times larger entire communication cost than the model trained for 20 epochs at each round. Hence, there is a trade-off between model performance of federated continual learning and communication efficiency, whereas FedWeIT variants consistently outperform (federated) continual learning baselines.

Table 5: Experimental results on the Overlapped-CIFAR-100 dataset with 20 tasks. All results are the mean accuracies over 5 clients, averaged over 3 individual trials.

| | Overlapped-CIFAR-100 with 20 tasks | | |
|---|---|---|---|
| **Methods** | **Accuracy** | **M Size** | **C2S/S2C Cost** |
| **FedProx** | 29.76 ± 0.39 | 0.061 GB | 1.22 / 1.22 GB |
| **FedProx-EWC** | 27.80 ± 0.58 | 0.061 GB | 1.22 / 1.22 GB |
| **FedProx-APD** | 43.80 ± 0.76 | 0.093 GB | 1.22 / 1.22 GB |
| **FedWeIT** | **46.78** ± 0.14 | 0.092 GB | **0.37 / 1.07** GB |

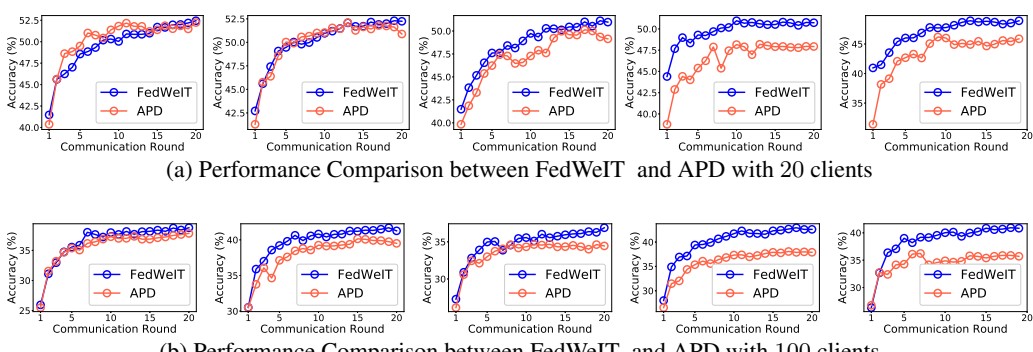

(a) Performance Comparison between FedWeIT and APD with 20 clients

(b) Performance Comparison between FedWeIT and APD with 100 clients

Figure 8: (a) Comparison of adaption for tasks between FedWeIT and APD with 20 clients in federated continual learning scenario (b) Comparison of adaptation for tasks between FedWeIT and APD with 100 clients in federated continual learning scenario. We visualize the last 5 tasks out of 10 tasks per client. *Overlapped-CIFAR-100* dataset are used after splitting instances according to the number of clients (20 and 100).

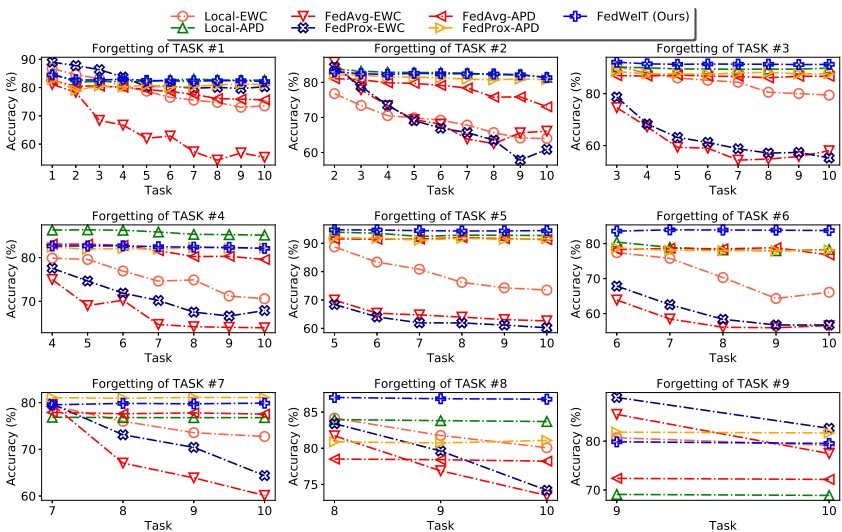

Figure 9: **Forgetting analysis.** Performance change over the increasing number of tasks for all tasks except the last task ($1^{st}$ to $9^{th}$) during federated continual learning on *NonIID-50*. We observe that our method does not suffer from task forgetting on any tasks.

## C.2  ABLATION STUDY FOR MODEL COMPONENTS

We perform an ablation study to analyze the role of each component of our FedWeIT. We compare the performance of four different variations of our model. **w/o B communication** describes the model that does not transfer the base parameter **B** and only communicates task-adaptive ones. **w/o A communication** is the model that does not communicate task-adaptive parameters. **w/o A** is the model which trains the model only with sparse transmission of local base parameter, and **w/o m** is the model without the sparse vector mask.

As shown in Table 6, without communicating **B** or **A**, the model yields significantly lower performance compared to the full model since they do not benefit from *inter-client knowledge transfer*. The model **w/o A** obtains very low performance due to catastrophic forgetting, and the model **w/o** sparse mask **m** achieves lower accuracy with larger capacity and cost, which demonstrates the importance of performing selective transmission.

Table 6: Ablation studies to analyze the effectiveness of parameter decomposition on WeIT. All experiments performed on NonIID-50 dataset.

| Methods | NonIID-50 | | |
|---|---|---|---|
| | Acc. | M Size | C2S/S2C Cost |
| **FedWeIT** | **84.11%** | **0.078** GB | **0.37 / 1.07** GB |
| w/o B comm. | 77.88% | 0.070 GB | 0.01 / 0.01 GB |
| w/o A comm. | 79.21% | 0.079 GB | 0.37 / 1.04 GB |
| w/o A | 65.66% | 0.061 GB | 0.37 / 1.04 GB |
| w/o m | 78.71% | 0.087 GB | 1.23 / 1.25 GB |

# D  DETAILED ALGORITHM FOR FEDWEIT

---

**Algorithm 2** Algorithm for FedWeIT

---

**input** Dataset $\{\mathcal{D}_c^{(1:t)}\}_{c=1}^C$, and Global Parameter $\boldsymbol{\theta}_G$
**output** $\{\mathbf{B}_c, \mathbf{m}_c^{(1:t)}, \boldsymbol{\alpha}_c^{(1:t)}, \mathbf{A}_c^{(1:t)}\}_{c=1}^C$
1: Initialize $\mathbf{B}_c$ to $\boldsymbol{\theta}_G$ for all $c \in \mathcal{C} \equiv \{1, ..., C\}$
2: **for** task $t = 1, 2, ...$ **do**
3:     **for** round $r = 1, 2, ..., R$ **do**
4:         Select communicable clients $\mathcal{C}^{(r)} \subseteq \mathcal{C}$
5:         **if** $r = 1$ **then**
6:             $\mathbf{A}_{c \in \mathcal{C}^{(r)}}^{(t-1,R)}$ and $\hat{\mathbf{B}}_{c \in \mathcal{C}^{(r)}}^{(t,r)}$ are transmitted from $\mathcal{C}^{(r)}$ to the central server
7:             Set a new knowledge base $kb^{(t-1)} = \{\mathbf{A}_j^{(t-1,R)}\}_{j \in \mathcal{C}^{(1)}}$
8:         **else**
9:             $\hat{\mathbf{B}}_{c \in \mathcal{C}^{(r)}}^{(t,r)}$ are transmitted from $\mathcal{C}^{(r)}$ to the central server
10:         **end if**
11:         Update $\boldsymbol{\theta}_G^{(r)} \leftarrow \frac{1}{|\mathcal{C}^{(r)}|} \sum_{c \in \mathcal{C}^{(r)}} \hat{\mathbf{B}}_c^{(t,r)}$
12:         Distribute $\boldsymbol{\theta}_G^{(r)}$ and $kb^{(t-1)}$ to client $c \in \mathcal{C}^{(r)}$ **if** $c_c$ meets $kb^{(t-1)}$ first, **otherwise** distribute only $\boldsymbol{\theta}_G^{(r)}$
13:         Minimize Eq. (2) for solving each local CL problems
14:     **end for**
15: **end for**

---

# E  FEDWEIT FOR ASYNCHRONOUS FEDERATED CONTINUAL LEARNING

---

**Algorithm 3** Algorithm for Asynchronous FedWeIT

---

**input** Dataset $\{\mathcal{D}_c^{(1:t)}\}_{c=1}^C$, and Global Parameter $\boldsymbol{\theta}_G$
**output** $\{\mathbf{B}_c, \mathbf{m}_c^{(1:t)}, \boldsymbol{\alpha}_c^{(1:t)}, \mathbf{A}_c^{(1:t)}\}_{c=1}^C$
1: Initialize $\mathbf{B}_c$ to $\boldsymbol{\theta}_G$ for all $c \in \mathcal{C} \equiv \{1, ..., C\}$
2: the knowledge base $kb \leftarrow \{\}$
3: **for** round $r = 1, 2, ...$ **do**
4:     **if** all clients finished the training **then**
5:         break
6:     **else**
7:         Select communicable clients $\mathcal{C}^{(r)} \subseteq \mathcal{C}$
8:         **for** $c \in \mathcal{C}^{(r)}$ **do**
9:             **if** new task $t'$ is arrived at the client $c_c$ **then**
10:                 Update the knowledge base $kb \leftarrow kb \cup \{\mathbf{A}_c^{(t'-1)}\}$ at the central server
11:             **end if**
12:             $\widehat{\mathbf{B}}_c^{(r)}$ are transmitted from the client $c_c$ to the central server
13:         **end for**
14:         Update $\boldsymbol{\theta}_G^{(r)} \leftarrow \frac{1}{|\mathcal{C}^{(r)}|} \sum_{c \in \mathcal{C}^{(r)}} \widehat{\mathbf{B}}_c^{(r)}$
15:         **if** a client $c_{c \in \mathcal{C}^{(r)}}$ still learn **then**
16:             Distribute $\boldsymbol{\theta}_G^{(r)}$ and $kb' \subseteq kb$ to the client $c_c$ **if** new task is arrived, **otherwise** distribute $\boldsymbol{\theta}_G^{(r)}$
17:             Minimize Eq. (2) for solving local CL problems at the client $c_c$
18:         **end if**
19:     **end if**
20: **end for**

---

We now consider FedWeIT under the asynchronous federated continual learning scenario, where there is no synchronization across clients for each task. This is a more realistic scenario since each task may require different training rounds to converge during federated continual learning. Here, *asynchronous* implies that each task requires different training costs (i.e., time, epochs, or rounds) for training. Under the asynchronous federated learning

Table 7: Averaged performance of FedWeIT with synchronous and asynchronous federated continual learning scenario.

| NonIID-50 | |
|---|---|
| **FedWeIT** | **Accuracy (%)** |
| **Synchronous** | $84.11 \pm 0.27$ |
| **Asynchronous** | $84.40 \pm 0.41$ |

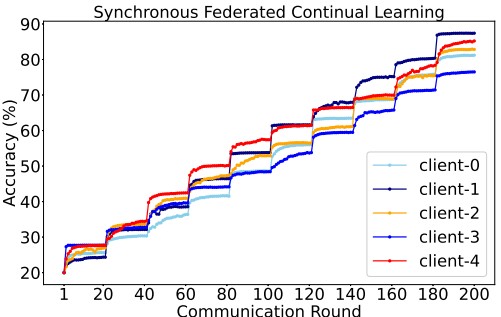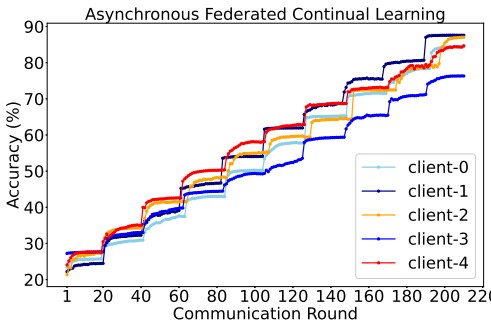

Figure 10: **FedWeIT with asynchronous federated continual learning** on *Non-iid 50* dataset. We measure the test accuracy of all tasks per client.

scenario, FedWeIT transfers any available task-adaptive parameters from the knowledge base ($kb'$) to each client. We provide the detailed algorithm in Algorithm 3. In Table 7 and Figure 10, we plot the average test accuracy over all tasks during synchronous / asynchronous federated continual learning. As shown in Figure 10, different tasks across clients at the same timestep require the same number of training rounds, receiving new tasks and task-adaptive parameters from the knowledge base simultaneously with the synchronous FedWeIT. On the other hand, with asynchronous FedWeIT, each task requires different training rounds and receives new tasks and task-adaptive parameters in an asynchronous manner. The results in Table 7 shows that the performance of asynchronous FedWeIT is almost similar to that of the synchronous FedWeIT.

