# OpenReview forum: "Federated Continual Learning with Weighted Inter-client Transfer"
_ICLR.cc/2021/Conference — Reject_

### Official Review · AnonReviewer1 · 2020-10-29
**Novel problem setting but writing is unclear and concerns in empirical results.**

**Rating:** 5
**Confidence:** 4

**Review:**

In this paper, the authors present a federated continual learning framework. By decomposing the local client parameter, the method could alleviate the effect of negative transfer and improve efficiency. Empirical results partly show the effectiveness of the proposed algorithm.

**Strength**
To my best knowledge, the problem setting is quite novel. Decomposing the parameter parts are also interesting.

**Weakness**
Writing is not good and many details are not clear. Some parts of the methodology lack explanation and the experiment results cannot support all claims of the authors. Below is my detailed opinion.

Equation 1 describes the decomposition.  However, some notations are not clearly defined. For example, what is L, I, O£¿ I have to guess that L is the depth of the network, I represent the input dimension, O represents the output dimension after I reading it many times.  I am not sure whether the above guess is right. Also, A_c^{(t)} is sparse task-adaptive parameters, this is not clear. I am not sure the sparse means in task-level or for every matrix A_c^{t} is sparse. I have to guess from equation 2, the author tries to concatenate all A, and perform sparse constrain.  I am not sure whether the above guess is right since the author did not explain it well.

In this paper, it seems that the author assumes that when a new task comes is knew ahead. In practice, it is hard to get such prior knowledge. Also, in the experiment part, it seems that different clients receive different tasks simultaneously, which is also quite not realistic. In practice, the new tasks could come asynchronously.

On the 2nd line of page 6, the author claims the efficiency of the algorithm is boosted since it requires |C| * (R * |B| + A). Isn't B is the same shape as \theta? I understand that the author assumes B is sparse, but we do not know how sparse it could be or can we recover the ground-truth sparse matrix or not. I can see that in the worst-case, without assumptions, this cannot improve efficiency. Empirical results only support that using the l1 constraint can output sparse results, while when this will work and correct is not clear in the methodology part.

Also, for this sparsity, lambda 1 is fixed across different tasks. This indicates different tasks have the same penalty. I do not see why this works since some tasks could be very similar to other tasks while some tasks are very distinct.  The same reason can be applied to lambda2 since the relationship between the current task and the previous task could be different. In figure 6, it seems that hyperparameters are very sensitive.
The plot of learned attention (Fig 5b).  I am not sure whether this is top-5 attention since there is no description (maybe I missed it). I assume that this is top-5 attention results. However, this cannot support the author that the proposed method can handle the negative transfer. First, empirically, why Traffic Sign has more weight than SVHN in the MNIST task? Moreover, This figure shows that empirically, using attention can focus on more important tasks/features and this should not be in the main contribution, and this is a well-known phenomenon. Has the ability to include the right task does not indicate excluding negative tasks.

In the middle plot of figure 6, I wish to see the whole task performance rather than selected tasks (this should be included in the appendix at least).

---

> ### Author Response · Authors · 2020-11-14
> **Initial Response 2**
>
> (9) The plot of learned attention (Fig 5b). this cannot support the author that the proposed method can handle the negative transfer. First, empirically, why Traffic Sign has more weight than SVHN in the MNIST task?
>
> - This makes sense since many traffic signs do **contain digits**(e.g., speed limit) and thus they are related to digit recognition tasks such as MNIST.
>
> ---------------------------
>
> (10) Figure 5 shows that empirically, using attention can focus on more important tasks/features and this should not be in the main contribution, and this is a well-known phenomenon. Has the ability to include the right task does not indicate excluding negative tasks.
>
> - Providing more attentions to relevant (or ‘right’) tasks and less attention on negative tasks, will minimize the effect of the negative tasks since their **relative weights will become very small**as more weights are given to relevant tasks. As shown in Figure 5 (b), MNIST-(0) task mainly receives knowledge transfer from relevant tasks such as Traffic sign-(5) task (42% over the total attention weights) and SVHN (1) task (33% over total attention weights). On the other hand, an irrelevant (negative) task, NotMNIST-(1), can hardly transfer the knowledge **(2% over the total attention weights)**.
>
> ---------------------------
>
> (11) In the middle plot of figure 6, I wish to see the whole task performance rather than selected tasks (this should be included in the appendix at least).
>
> - We have included plots of all tasks in the **appendix of the revision**as suggested (Please see Figure 9 in appendix).  Also, further note that our FedWeIT obtains the best BWT score and accuracy compared to baselines as shown in Table 1, Figure 4, and 6.

---

> ### Author Response · Authors · 2020-11-14
> **Initial Response 1**
>
> We appreciate your constructive feedback. We address your comments below:
>
>
> (1) Writing is not good and many details are not clear.
>
> - We did our best to clarify the details you find unclear in the **revision**. Please also note that other reviewers find the our paper well-written ( “the paper is well presented and organized” (R2), “The paper is generally well-written” (R4)).
>
> ---------------------
>
> (2) Some notations are not clearly defined.
>
> - Due to the page limit, we have suppressed definitions of several typical expressions. However, reflecting your comment, we have **clearly defined them back**in the revision (Please see Page 5).
>
> -------------------------------
>
> (3) A_c^{(t)} is sparse task-adaptive parameters, this is not clear.
>
> - As described in the paper, $A_c^{(t)}$ is the task-adaptive parameter of the task $t$ arriving at the client $c$. Thus, “sparse task-adaptive parameter” means that **$A_c^{(t)}$ is sparse**, containing mostly zero elements. In equation 2, we clearly describe that the sparsity-inducing regularization term makes $A_c^{(t)}$ sparse. We have updated the description in the **revision**for better clarity (Please see Page 5).
>
> --------------------
>
> (4) Different clients receive different tasks simultaneously, which is also quite not realistic. In practice, the new tasks could come asynchronously.
>
> - Although our scenario assumes synchronous federated continual learning that $t^{th}$ tasks from different clients ($\mathcal{T}^{(t)}_{1:c}$), our FedWeIT can easy extend to asynchronous federated continual learning scenarios using simple modifications. We have included a modified algorithm for asynchronous federated continual learning in the **appendix of the revision**. We have performed **more experiments** under this asynchronous setting. Please see Table 7 and Figure 10 in the appendix of the revision. The result of Table 7  is as follow:
>
>
> |FedWeIT for | Non-IID dataset |
> |:-------------------------:|:------------------------------:|
> | Scenario                |                 Accuracy (%)           |
> | **Synchronous**|           84.11 ± 0.27          |
> | **Asynchronous**|           84.40 ± 0.41        |
>
> We observe that the asynchronous FedWeITperforms as well as the synchronous FedWeIT in the original paper. We thank you again for the helpful suggestion.
>
> --------------------------
>
> (5) We do not know how sparse the parameters could be. Empirical results only support that using the l1 constraint can output sparse results, while when this will work and correct is not clear in the methodology part.
>
> - Increasing sparsity can be done by increasing $\lambda_1$ for the **sparsity regularization in Eq. (2)**. Further, we empirically show that FedWeIT shows significant performance/efficiency gain using **30% and 3%** of the parameters for $\hat B$ and $A$ compared to the dense models, respectively, as described in the paragraph of “Efficiency of FedWeIT” on page 7. Moreover, we provide experimental results with different sparsity rates in Figure 5 (a) and extensive experiments on 8 different datasets (CIFAR-100, CIFAR-10, MNIST, FashionMNIST, Traffic Sign, NotMNIST, FaceScrub, and SVHN) and 2 neural networks (LeNet, ResNet-18).  Our FedWeIT consistently outperforms baselines with high sparsity rates in all settings we validate on.
>
> ---------------------------
>
> (6) lambda 1 and lambda 2 is fixed across different tasks. This indicates different tasks have the same penalty.
>
> - Task-specific hyperparameter tuning will obviously yield further performance gain but will be costly. Thus we set task-general hyperparameter during training for the **ease of use**, since our method is **not sensitive to hyperparameters**.
>
> ---
>
> (7)  I do not see why this works since some tasks could be very similar to other tasks while some tasks are very distinct.
>
> - Moreover, our model **does consider the task relevance**by learning the masking variable $m_c^{(t)}$ and the weights on the task-adaptive parameters $\alpha_{i,j}^{(t)}$ in Eq.(1). The masking variable confines the effect of the task $t$ on the shared parameters, to the ones that are masked by $m_c^{(t)}$, and the learned weights determines which of the task-adaptive parameters are more useful for the given task.
>
> ---------------------
>
> (8) In figure 6, it seems that hyperparameters are very sensitive.
>
> - Our FedWeIT is **not sensitive**to small change in the hyperparameters. Note that the parameters we consider in Figure 6 are (multiple) orders of magnitudes different (10x for $\lambda_1$ and 3,000x for $\lambda_2$) as shown in Figure 6 Left. FedWeIT generally performs well on diverse tasks with task-general pre-defined hyperparameters. However, the hyperparameters are essential for controlling the trade-off between accuracy over communication cost.

---

> > ### Comment · AnonReviewer1 · 2020-11-23
> > **Feedback on response.**
> >
> > Thanks for your response. After reading your response and other people's review, I think they solved most of the problems in writing and experiments. However, some explanation about sparsity and $\lambda_1$ and $\lambda_2$ still did not convince me. I will raise my score from 4 to 5.
> >
> > Asking about why fixing lambda_1 and lambda_2 is not a concern for the empirical results and hyperparameter sensitivity. My question is why using fixed lambda_1 and lambda_2 works, considering tasks are very different?  The explanation that saying our method is not hyperparameter sensitive does not answer the question.  Also, as the author mentioned, the sparseness results are only empirical showed. So, there is no guarantee that |C| * |R|* |\theta| is worse than |C| × (R × |\theta| + (|C| − 1) × |A|) . It could be possible that |A| and \theta are either not sparse or the sparseness will dramatically decrease the performance. I have no question about your empirical results. However, to claim the better space complexity or communication efficiency, it requires deeper analysis.

---

> > > ### Author Response · Authors · 2020-11-23
> > > **Response on the hyperparameters lambda_1 and lambda_2**
> > >
> > > We sincerely thank you for the feedback.
> > >
> > > - Please refer to the **answer (7)** in the response above. You seem to have misunderstood that the hyperparameters $\lambda_1$ and $\lambda_2$ are the terms that control the knowledge transfer across tasks. However, they are not. The terms that control the amount of knowledge transfer across tasks, are **learnable weights $\alpha_{i,j}^{(t)}$** in Eq.(1), and $\lambda_2$ simply controls the **scale of the knowledge transfer term** in Eq.(2). Please see Eq.(1)  of the paper as well. Thus the hyperparameters $\lambda_1$ and $\lambda_2$ need not be tuned per task.
> > >
> > > - Please also note that $\lambda_1$ is a **hyperparameter for sparsity-inducing regularization ($\ell_1$-regularization)**, and not the one that controls the effect of the knowledge transfer.
> > >
> > > Please let us know if this has clarified your question on the use of the same $\lambda_1$ and $\lambda_2$ across tasks.

---

> > > > ### Author Response · Authors · 2020-11-25
> > > > **Clarification on Lambda_1 and Lambda_2**
> > > >
> > > > Dear reviewer,
> > > > Could you check the response above since we only have around 10 hours until the end of the interactive discussion phase?To summarize, basically, the task relevance is considered by the **learnable weights $\alpha_{i,j}^{(t)}$** (Please see 7 in the original response as well) and $\lambda_2$ simply controls the **scale of the knowledge transfer term with respect to other terms** in Eq.(2). We hope that this satisfactorily clarifies your confusion.

---

> > > > > ### Comment · AnonReviewer1 · 2020-11-25
> > > > > **Feedback and technical concern on sparsity**
> > > > >
> > > > > Thanks for the clarification.
> > > > >
> > > > > I understand that increasing lambda makes the matrix sparse. I am asking the solution is possible to be either not sparse or the sparseness will dramatically decrease the performance. Varying lambda can make the matrix more sparse does not imply anything about the performance. For example, what if your network size is small? This situation is possible since local client memory is limited. In that case, your matrix size $A$ will also be small. In that case, can you also guarantee that making the $A$ be sparse and meanwhile also preserving good performance?
> > > > >
> > > > > Another technical issue in the paper is that I just realized that you did not mention how you optimize the l1 regularization (subgradient or soft thresholding). Considering most existing tools such as PyTorch/TensorFlow use subgradient to solve non-differential objectives, it could generate non-sparse results since subgradient has poor convergence properties for non-smooth functions.
> > > > >
> > > > > For lambda1 and lambda2, I fully understand that lambda1 controls sparse and lambda2 controls catastrophic forgetting. Unfortunately, your answer cannot convince me that these two should be consistent across different tasks.

---

> > > > > > ### Author Response · Authors · 2020-11-25
> > > > > > **Regarding $\lambda_1$ and $\lambda_2$**
> > > > > >
> > > > > > **For lambda1 and lambda2, I fully understand that lambda1 controls sparse and lambda2 controls catastrophic forgetting. Unfortunately, your answer cannot convince me that these two should be consistent across different tasks.**
> > > > > >
> > > > > > - Please note that the **task relevance is considered by the learnable weights $\alpha_{i,j}^{(t)}$, not only by $\lambda_2$**. Thus, even if **$\lambda_2$ is set to be the same**, the amount of the knowledge transfer from one task to another, **will not be the same**.
> > > > > >
> > > > > > - As for $\lambda_1$, adjusting it differently for each task will be the most optimal, as you mentioned, and **we agree with that**. However, doing so will be extremely difficult in practice since it requires to **search for a large combination of hyperparameters**. Note that there are $10$ clients, each of which comes with $10$ tasks. Thus, we need to search for the optimal combination of **$100$ different $\lambda**for 100 different tasks if we individually adjust the $\lambda_1$, in our experimental setting, which is highly impractical.
> > > > > >
> > > > > > - We would like to kindly remind you that our focus is **not obtaining an optimal model in terms of sparsity-performance tradeoff**, but a model that can effectively solve the **inter-task interference**problem, which is newly introduced with the novel problem of **federated continual learning**, while **reducing** the communication cost but not optimizing it. We have also shown that our model works well under various settings, and obtaining optimal sparsity parameters $\lambda_1$ will be beyond the scope of this paper, and as it is a general problem that does not only apply to our setting.

---

> > > > > > ### Author Response · Authors · 2020-11-25
> > > > > > **Sparsity guarantee of subgradients**
> > > > > >
> > > > > > **Another technical issue in the paper is that I just realized that you did not mention how you optimize the l1 regularization (subgradient or soft thresholding). Considering most existing tools such as PyTorch/TensorFlow use subgradient to solve non-differential objectives, it could generate non-sparse results since subgradient has poor convergence properties for non-smooth functions.**
> > > > > >
> > > > > > - We used the $\ell_1$ regularization term in the Tensorflow objective, but it did actually produce sparse models that made the communication of the parameters **more** efficient in practice. Please see the results in Table 1, Figure 4, and Figure 5 of the paper.
> > > > > >
> > > > > > - However, we agree with the poor convergence of subgradient methods and we can use proximal gradients (with iterative soft-thresholding as the proximal operator), or Dual averaging methods [Xiao et al. 11] that have better convergence guarantee. Yet which method to use in order to sparsity the parameters is not part of our framework, but is rather an implementation detail. Please also note that the focus of this paper is not obtaining a model with **optimal sparsity**. The focus is on the novel **inter-task knowledge transfer** for a novel **federated continual learning problem**, in a communication-efficient manner via sparse parameter communication.
> > > > > >
> > > > > > [Xiao et al. 11] Dual Averaging Methods for Regularized Stochastic Learning and Online Optimization, JMLR 2011

---

> > > > > > ### Author Response · Authors · 2020-11-25
> > > > > > **Sparsity vs. Performance tradeoff.**
> > > > > >
> > > > > > **I am asking the solution is possible to be either not sparse or the sparseness will dramatically decrease the performance. Varying lambda can make the matrix more sparse does not imply anything about the performance. For example, what if your network size is small? This situation is possible since local client memory is limited. In that case, your matrix size  will also be small. In that case, can you also guarantee that making the  be sparse and meanwhile also preserving good performance?**
> > > > > >
> > > > > > - We report the **performance of the model with varying degree of sparsity in Figure 5(a)**, which shows that it is possible to obtain a model that performs as well as the full model, while reducing the communication cost as much as 40% of the full model.
> > > > > >
> > > > > > - We report the results with **both small models (LeNet, Table 1, Figure 4, and Figure 5) and large models (ResNet-18, Table 2)**, which show that we can obtain sparsity without much compromise of the accuracy. Since our method is **already outperforming** baselines, having **less than optimal**improvements will be less important, although finding an optimal sparsity-performance tradeoff may bring further improvements over the already large performance gains our method achieves.
> > > > > >
> > > > > > We **fully agree**that it would be good to have some theoretical analysis and guarantee on sparsity, but please understand that this is not the focus of this paper. We hope you understand that our focus is on the proposal of a **novel continual learning problem in a federated learning setting**, and a novel algorithm to prevent **inter-task interference**which is a new challenge in this unique setting. We have empirically shown that the model largely outperforms existing methods, and is communication efficient with both small and large networks.
> > > > > >
> > > > > > We thank you for your insightful comments. Please let us know if there is anything else we should clarify.

---

> > > ### Author Response · Authors · 2020-11-23
> > > **Response to your comments "sparseness is only empirically shown"**
> > >
> > > - Please note that the sparsity of $|A|$ can be controlled by the hyperparameter $\lambda_1$ in Eq.(2), and $\Omega$ is a sparsity-inducing regularizer, which in this case is a $\ell_1$-regularization. Thus **increasing $\lambda_1$ will make $|A|$ to be sparser**, and the trade-off between the sparsity and the performance can be controlled, by adjusting $\lambda_1$.
> > >
> > > - Please also note that $\theta$ needs not be sparse since this is the final parameter that is being used at each client. What is communicated instead, is the difference of $\hat{B}_c^{(t)} = B_c^{(t)} \odot {m_c^{(t)}}$ across consecutive iterations, which will be trivially **sparse since the mask parameter $m_c^{(t)}$ is sparse** (Please see Eq.(1), Eq.(2), and Algorithm 1).
> > >
> > > - While providing sparsity guarantee may be beneficial, most of the methods on neural network pruning or model compression do not provide such guarantees either. Also, please note that our work is **not proposing a communication-efficient federated learning algorithm for a general setting**, but is rather proposing **a new problem setting of federated continual learning**as well as a **framework to effectively handle inter-task knowledge communication while avoiding inter-task interference**, without excessive communication cost. We also believe that the experimental results we provide shows sufficient evidence that the model is communication efficient in practice. However your suggestion is helpful and we will consider it as future work.

---

> ### Author Response · Authors · 2020-11-23
> **Responses and Revision uploaded**
>
> Dear reviewer,
>
> Could you check our responses to your comments as well as the revision that reflects them?
> We have clearly described all the notations, clarified the points you find unclear, provided the results on the asynchronous continual learning setting you suggested, and included the forgetting curve for all tasks (except the last task).
> We would like your feedback since we cannot interact with you after this Tuesday, which is the end of the interactive discussion phase.
>
> Thanks,
> Authors

---

### Official Review · AnonReviewer3 · 2020-11-01

**Rating:** 7
**Confidence:** 3

**Review:**

The authors propose a federated continual learning setting where each node has a non-iid stream and a different dataset. The authors address this by extending the parameter decomposition method of Yoon et al

Strengths:

- The authors introduced a relevant new task that introduces concepts from continual learning to federated learning. The task setting seems practical overall as different nodes will often be following a different distribution
-The algorithm proposed is interesting. I would like further discussion on the differences to Yoon et al
-The results are good and evaluate a lot of  relevant factors

Weakness:

- Although the setting proposed is interesting, certain aspects of it seem artificial: although it seems very relevant that each node is a different dataset, strong non-iid behavior per node seems not realistic for many settings
-What happens when some nodes start learning much earlier than others
- A discussion of challenges in these settings and other potential methods
- Results are shown for very simple LeNet architecture only
- The algorithm proposed is interesting, but more motivation and other possible alternatives in this setting would improve the paper

---

> ### Author Response · Authors · 2020-11-14
> **Initial Response**
>
> We appreciate your constructive feedback. We address your comments below:
>
>
> (1) The algorithm proposed is interesting. I would like further discussion on the differences to Yoon et al
>
> - Our work is novel over (Yoon et al, 2020) in the following aspects:
>
> - 1) We propose a novel **federated continual learning problem**, where each client learns on a sequence of tasks while **sharing the task-specific knowledge**across clients. Moreover, we highlight a novel challenge with this new problem, which we refer to as **inter-client knowledge transfer**, where irrelevant task knowledge from other clients negatively affects the model's performance on the given task when we naively apply existing federated learning algorithms to the problem.
>
> - 2) To this end, we propose a novel **inter-client knowledge transfer**, which allows to selectively transfer only the knowledge from only the relevant tasks learned at other clients, when the local model at each client is learning for a new task, in order to minimize **inter-task interference**. We also propose a communication-efficient algorithm to reduce the communication overhead in transmitting the task-adaptive parameters between the server and the client.
>
> - 3) We show through experiments that **a naive combination of APD (Yoon et al, 2020) with an existing FL (FedProx-APD) significantly underperforms** our method and is communication inefficient (Please see Table 1 and Figure 4).
> ----------------
>
> (2-1) Although the setting proposed is interesting, certain aspects of it seem artificial: although it seems very relevant that each node is a different dataset, strong non-iid behavior per node seems not realistic for many settings.
>
> - We consider both cases (strong non-iid / not strong non-iid) of the dataset for our evaluation. While Non-IID 50 dataset is a strong non-iid, Overlapped-CIFAR-100 dataset has some overlapping (**relevant**) tasks. As described on page 6, for Overlapped-CIFAR-100 dataset, we split CIFAR-100 dataset into 20 non-iid superclasses tasks. And we randomly sample instances of a task to create multiple different tasks. Overlapped-CIFAR-100 dataset thus has overlapped classes (**very relevant**) but not with a duplication of the instance.
>
> ---
>
> (2-2) What happens when some nodes start learning much earlier than others?
>
> - We thank you for the insightful comment. We agree that different tasks may require different amount of time to train, and thus it makes more sense to consider an **asynchronous federated continual learning**. Fortunately, our method can easily extend to an asynchronous algorithm for such a setting, with simple modification of the algorithm. Basically, we can tackle the scenario by allowing each client to receive any task-adaptive parameters from the knowledge base that are available when the client initializes training on the new task. We have included a modified pseudo-code of the algorithm for asynchronous FedWeIT in Algorithm 3 of the Appendix, in the revised version of the paper, and performed **additional experiments under the asynchronous FCL settings** (Table 7 and Figure 10 in the Appendix). The results are as follows:
>
> |FedWeIT for | Non-IID dataset |
> |:-------------------------:|:------------------------------:|
> | Scenario                |                 Accuracy (%)           |
> | **Synchronous**|           84.11 ± 0.27          |
> | **Asynchronous**|           84.40 ± 0.41        |
>
> We observe that the **asynchronous FedWeIT**performs as well as the synchronous FedWeIT in the original paper. We thank you again for the helpful suggestion.
>
> -----------------
>
> (3) A discussion of challenges in these settings and other potential methods. Results are shown for very simple LeNet architecture only
>
> - You may have missed our **results with ResNet-18 in Table 2**. Our FedWeIT still significantly outperforms the baselines with the ResNet-18 architecture, which is expected since the algorithm is agnostic to the choice of the backbone network.

---

### Official Review · AnonReviewer2 · 2020-11-01
**Official Blind Review #2**

**Rating:** 6
**Confidence:** 5

**Review:**

This paper investigates a new problem – federated continual learning by Federated Weighted Inter-client Transfer. The key idea is to decompose the network weights into global federated parameters and sparse task-specific parameters such that each client can selectively receive knowledge from other clients by taking a weighted combination of their task-specific parameters. The experiment results in two contrived datasets demonstrate the effectiveness of the proposed method.

Strength:
+ This paper is well presented and organized.
+ The proposed federated continual learning framework is innovative and technically sound.
+  The experiment results are solid.

Weakness:
- The optimization procedure for Eq. (2) is not provided.
- Some details are missing (as shown below).

The following are some questions that I concern.

1.	The authors address the importance of federated continual learning from the aspect of continual learning, but is there any difference between federated continual learning and federated learning considering that most existing federated learning methods (fedavg and fedprox) are agnostic of client id?
2.	How to train alpha in Eq. (1)? Is this a learnable parameter with sigmoid activation? If yes, how to set the parameters for testing on different tasks? Moreover, the whole testing process is confusing to me.
3.	The detailed optimization procedure for Eq. (1) is not provided.
4.	For the training objective Eq. 2, intuitively, authors propose decomposable parameters and want Base parameters B to be sparse with the help of task adaptive parameters A.  However, it is interesting to see that there is no constraint to encourage B and A to focus on different aspects. I thus doubt that the communication efficiency brought by the sparse parameters mainly benefits from the sparsity constraints, i.e., 2nd term in Eq (2), but not the decomposable parameters. The authors are encouraged to show that the base parameters (m*B) are more sparse than the model trained with existing federated methods with similar sparsity constraints.  Otherwise, the communication efficiency of the proposed method would not stand.
5.	 For the third term in Eq. (2), whether it is necessary to impose the constraint on  A_c^i (i<|t|), as in Eq. (1)?

---

> ### Author Response · Authors · 2020-11-18
> **Initial Response**
>
> We appreciate your constructive feedback. We address your comments below:
>
> **(1) Is there any difference between federated continual learning and federated learning considering that most existing federated learning methods (fedavg and fedprox) are agnostic of client id?**
>
> - In the federated continual learning scenario, the clients should learn on heterogeneous sequence of tasks, rather than on a single task as done in conventional federated learning cases. This poses a new challenge in how to **communicate**and **aggregate** knowledge across clients, since a task at a particular client may or may not be useful for the learning of the model on another client, on a specific task (Please see Figure 1(a)).
>
> - In the introduction (Figure 1(b)), we have shown that **simple aggregation of learned knowledge**may introduce inter-client interference, where the aggregation of the parameters across clients results in performance degeneration, due to incompatibility of the tasks. We have also shown through experiments that baselines that naively combine federated learning algorithms with continual learning models (FedProx-EWC and FedProx-APD in Table 1), are highly suboptimal.
>
> - Please let us know if this does not address your comment, since we may have misunderstood your intention.
>
> -------------
> **(2) How to train alpha in Eq. (1)? Is this a learnable parameter with sigmoid activation? If yes, how to set the parameters for testing on different tasks? The detailed optimization procedure for Eq. (1) is not provided.**
>
> - $\alpha^{(t)}$ is indeed a **learnable parameter**for task $t$. We initialize the attention parameter as sum to one, $\alpha^{(t)}_{c,j}\leftarrow 1/|\alpha^{(t)}_c|$, and then optimize them as free variables.
>
> - We have clarified the training procedure in the revision (Please see Section B in the appendix of the revision). The trainable parameters at each client are optimized using equation (2) during training, with the Adam optimizer as described in the Section B in the appendix.
>
> - In the testing phase, we make predictions on each sample from a given task with the task-specific model for each task, whose parameters are obtained by summing up the task-shared and task-adaptive parameters. For test, we measure the performance after the completion of all learning phases (i.e., all tasks) over 3 individual trials.
>
> ----------------
> **(3-1) For the training objective Eq. 2, it is interesting to see that there is no constraint to encourage B and A to focus on different aspects.**
>
> - Since the base parameter $B_c$ is shared across clients and heterogeneous tasks, $B$ will learn task-general knowledge. On the other hand, since the task-adaptive parameter $A^{(t)}_c$ captures the knowledge that is not captured by $B$ for each task $t$ as done in residual learning, it will learn task-specific knowledge.
> - However, as you mentioned, we do not have an explicit constraint in encouraging $B$ and $A^{(t)}_c$ to focus on different aspects. This is an insightful comment, and we believe that we can use recent approaches [1] and [2], to perform orthogonal updates for the two types of parameters.
>
> [1] Farajtabar, Mehrdad, et al. "Orthogonal gradient descent for continual learning." International Conference on Artificial Intelligence and Statistics (AISTATS). PMLR, 2020.
> [2] Chaudhry, Arslan, et al. "Continual Learning in Low-rank Orthogonal Subspaces." Advances in Neural Information Processing Systems (NeurIPS), 2020.
> ---
>
> **(3-2) I thus doubt that the communication efficiency brought by the sparse parameters mainly benefits from the sparsity constraints, but not the decomposable parameters.**
> - Parameter decomposition of FedWeIT has a crucial role in reducing communication cost of FedWeIT. The main reason why the highly sparse task-adaptive parameters ($A^{(t)}_c$) work well, is because the shared dense parameter $B$ captures most knowledge, and task-adaptive parameters only need to capture the residual of the knowledge captured by $B$.
>
> ---
>
> **(3-3) The authors are encouraged to show that the base parameters** ($m*B$) **are more sparse than the model trained with existing federated methods with similar sparsity constraints.**
> - The original paper (and the revision) contains the experiments with sparse communication of the base parameters ($m*B$) in **Figure 5 (a)**. Models with different communication costs are obtained by controlling the sparsity of $m*B$, and the results show that our FedWeIT obtains superior performance with only a fraction of communication cost of the baselines'.
>
> ---------------------
> (4) For the third term in Eq. (2), is it necessary to impose the constraint on $A_c^i (i<|t|)$, as in Eq. (1)?
>
> - The third term minimize the catastrophic forgetting of FedWeIT by updating the task-adaptive parameters to reflect the change of the base parameter, such that it maintains the original solutions for the target tasks. Therefore, we impose the constraints on all $i$, where $i = 1, ..., t-1$.

---

> > ### Comment · AnonReviewer2 · 2020-11-24
> > **Feedback to Initial Response**
> >
> > The authors have addressed most of my previous concerns. However, regarding (3-3), the baseline methods including FedAvg-APD/FedProx-APD should be also trained with the sparsity constraint over base parameters.
> >
> > In Eq. 2, the authors impose l1 constraint on mask m, while the baseline methods (FedAvg-APD) are trained without any constraints on the mask (Eq. 3 in APD). In my opinion, the sparsity constraint is the main reason for the significant reduction of C2S cost for the proposed methods.
> >
> > Since it is simple to impose the constraint on all baseline methods, including even the fedprox, it is thus necessary for the authors to make a comparison.

---

> > > ### Author Response · Authors · 2020-11-24
> > > **Response**
> > >
> > > Dear Reviewer,
> > >
> > > We thank you for going over our responses, and find them satisfactory.
> > > Regarding applying the sparse masks to baselines (FedAvg-APD, FedProx-APD), please note that we **do apply sparse masks $m$, trained with the $\ell_1$-regularization**on the shared parameter $B$ of the baselines, for the results in **Figure 5(a)**. That is why we could reduce the communication costs for the baselines as well. Thus we are already making the comparison that you have suggested, and we will make it more clear in the caption of Figure 5, in the revision.
> > >
> > > We thank you again for your efforts and helpful suggestions. Please let us know if you have any other concerns, as we will do our best to address them.

---

> > > ### Author Response · Authors · 2020-11-24
> > > **Please see the code of the Supplementary Material**
> > >
> > > In the supplementary material, we provide the code of our FedWeIT and APD baselines.
> > >
> > >
> > > Please see **./models/apd/apd_local.py L264-L267** in the code, FedAvg-APD and FedProx-APD apply sparse masks $m$, trained with the $\ell_1$-regularization on the shared parameter if $\textit{sparse-communication}$ option is $\textit{True}$ for experiments in **Figure 5 (a)**.

---

### Official Review · AnonReviewer4 · 2020-11-08
**This paper proposes a method to solve continual federated learning. It allows the models to leverage task specific information by other peers while preventing the negative interactions. The paper is tackling a cool problem but is borderline due to contributions and experiments.**

**Rating:** 6
**Confidence:** 5

**Review:**

Combination of federated learning and continual learning is a timely problem. Given the trending popularity of either of them it's the time to tackle this problem. The problem that is tackled is nice but the proposed formulation looks incremental.

The paper is generally well written but has a few typos along the paper like:
Page 2: "... need to selective utilize .."
Page 2: ... once when ..."
Page 7: "dtaset"

In section 3.1 the relation between tasks of the clients is not clear. Are the tasks at step i of all tasks related to each other?

In equation 1: In the third term --> Isn't there a case for transfer between the task of the same client to future tasks?

Section 3.3 for training: Imagine we are at task t. Then, the minimization problem in equation 3 involves solving for A^i. Why do one need to find A^i for a previous task and find a new parameter for that? Isn't that task already gone?Although, does this mean that the parameters to be estimated at each step is growing with the number of tasks?

The NonIID data set is a cool combination of different smaller datasets.

 The experiments look overall good. However, the part that shows alleveriating catastrophic forgetting seems rather less elaborated. Why only 6th and 8th tasks? Why not demonstrate forgetting of task 1 over the time. Why not all tasks? And, Fig. 6 only shows up to 5 tasks. How many consecutive tasks does the proposed method handle?

This works look like an increment on the top of (Yoon et al, 2020). Like making their approach adapted to a federated learning case. Thus, the contribution of the works is rather limited.

---

> ### Author Response · Authors · 2020-11-14
> **Intial Response**
>
> We appreciate your constructive feedback. We address your comments below:
>
> (1) In section 3.1 the relation between tasks of the clients is not clear. Are the tasks at step $i$ of all tasks related to each other?
>
> - There is **no relationship** among the task $\mathcal{T}^{(t)}$ received at step $t$, across clients. We have clarified this in the revision.
>
> -----------------
> (2) In equation 1: In the third term --> Isn't there a case for transfer between the task of the same client to future tasks?
>
> - As described on page 5, the third term in equation 1 describes the knowledge transfer transmitted from other clients. Separately, the model **transfers the knowledge** from the previous tasks learned from the same clients ($\mathcal{T}^{(1:t-1)}_c$) using the final regularization term in equation 2. We will provide more details about the term in the response to the next question.
> -----------------
>
> (3) Section 3.3 for training: Why does one need to find $A^{(i)}$ for a previous task and find a new parameter for that?
>
> - As described on page 5, the third term in equation 2 prevents catastrophic forgetting for previous tasks. The model can not access the data of previous tasks but model parameters can be reconstructed using (current state) base parameters ($B^{(t)}$) and task-adaptive parameters ($A^{(i)}$). At that time, the base parameter $B^{(t)}$ is already updated by the arriving tasks, which may lead to semantic drift. Thus the third term prevents forgetting by updating $A^{(i)}$ from the previous tasks to reflect the change of base parameters $B$, to maintain the original solution on previous tasks.
> ----------------
>
>
> (4) Does this mean that the parameters to be estimated at each step is growing with the number of tasks?
>
> - When new task $t$ arrives, a client $c$ of FedWeIT newly generates task-adaptive parameters $A^{(t)}_c$ to learn task-adaptive knowledge. Hence, the total capacity of the model is increased during training. However, as described on page 5, highly sparsified task-adaptive parameters require $2$-$3$% capacity compared to original model parameters, which is marginal.
>
> ----------------
> (5) In figure 6, why only 6th and 8th tasks? Why not demonstrate forgetting of task 1 over time. How many consecutive tasks does the proposed method handle?
>
> - We selected random tasks, but have added plots of all given tasks in the appendix of the revision (Please see Figure 9 in the Appendix). Also, note that our model obviously shows the best BWT score and accuracy compared to baselines as shown in Table1 and Figure 6 right. Furthermore, our FedWeIT is applicable to the larger number of tasks per client. We have included 20 tasks experiments on the Overlapped-CIFAR-100 dataset as below (Please see Table 5 in the Appendix).
>
> |Overlapped-CIFAR-100 | Dataset with | 20 Tasks                             ||
> |:-------------------------:|:------------------------------:|:----------------------:|:----------------------:|
> | Methods                 |                 Accuracy (%)           |         Model Size       |         C2S / S2C Cost|
> | FedProx-EWC|           27.80 ± 0.58          |     0.061 GB    | 1.22 / 1.22 GB |
> | FedProx-APD|           43.80 ± 0.76        |      0.093 GB    |  1.22 / 1.22 GB |
> | **FedWeIT (Ours)**   |           **46.78 ± 0.14**         |    **0.092 GB**    |  **0.37 / 1.07 GB**     |
>
> ----------------------
> (6) This works look like an increment on the top of (Yoon et al, 2020). Like making their approach adapted to a federated learning case. Thus, the contribution of the works is rather limited.
>
> Our work is not incremental but is highly novel over (Yoon et al, 2020) in the following aspects:
>
> - 1) We propose a novel **federated continual learning problem**, where each client learns on a sequence of tasks while **sharing the task-specific knowledge**across clients. Moreover, we highlight a novel challenge with this new problem, which we refer to as **inter-client knowledge transfer**, where irrelevant task knowledge from other clients negatively affects the model's performance on the given task when we naively apply existing federated learning algorithms to the problem.
>
> - 2) To this end, we propose a novel **inter-client knowledge transfer**, which allows to selectively transfer only the knowledge from only the relevant tasks learned at other clients, when the local model at each client is learning for a new task, in order to minimize **inter-task interference**. We also propose a communication-efficient algorithm to reduce the communication overhead in transmitting the task-adaptive parameters between the server and the client.
>
> - 3) We show through experiments that **a naive combination of APD (Yoon et al, 2020) with an existing FL (FedProx-APD) significantly underperforms** our method and is communication inefficient (Please see Table 1 and Figure 4).

---

> ### Author Response · Authors · 2020-11-23
> **Responses and the revision uploaded**
>
> Dear reviewer,
>
> Could you check our responses to your comments as well as the revision that reflects them?
> We have answered all your questions, provided the experimental results you have requested **(20 tasks experiments and the task forgetting curve for all tasks)**, and clarified your concern regarding the **novelty over [Yoon et al. 2020]**
>
> Thanks,
> Authors

---

> ### Author Response · Authors · 2020-11-25
> **The end of the interactive discussion period in less than 10 hours**
>
> Dear Reviewer,
>
> Could you check the response and the revision since the interactive discussion phase will end less than 10 hours?
> We have provided the two experimental results you requested, provided clarification on your questions, and clarified on the novelty over [Yoon et al. 20]. Please let us know if there is anything else we should provide. We thank you for your efforts in reviewing our paper, as well as the insightful and constructive comments.

---

### Author Response · Authors · 2020-11-18
**Summary of the Initial Revision**

We thank the reviewers for constructive comments. We appreciate that the reviewers consider the tackled problem to be interesting (R2, R3, R4), innovative (R2),  and practical (R3), and the solution to be novel (R1, R4) with solid results (R2, R3). The reviewers also mention that the paper is well-written (R2, R3, R4). **We have highlighted the updates in purple (Please see the revised version of the paper)**:

-------------------

(1) Line 4 on page 4,  "Please note that there is no relation among the tasks $\mathcal{T}^{(t)}_{1:c}$ received at step $t$, across clients." (**R4**)

------------

(2) Line 2-4 on page 5, "Here, $L$ is the number of the layer in the neural network, and $I_l, O_l$ are input and output dimension of
the weights at layer $l$, respectively." (**R1**)

------------------

(3) After Equation (2) on page 5, "$\Omega(\cdot)$ is a sparsity-inducing regularization term for all task-adaptive parameters." (**R1**)

-----------------

(4) Section B on page 12, "We initialize the attention parameter as sum to one, $\alpha^{(t)}_{c,j}\leftarrow 1/|\alpha^{(t)}_c|$." (**R2**)

--------

(5) Table 5 on page 13, **Experiments on FCL with 20 tasks**. (**R4**)
- We have performed additional experiments on federated continual learning with a **larger number of tasks (i.e., 20 tasks)**, and have included the experimental results in **Table 5 of the Appendix.**

-----------------

(6) Figure 9 on page 14, **Experiments on catastrophic forgetting for all tasks**. (**R1** and **R4**)
- We have included **plots that show the performance change of all tasks** during the course of continual learning, in Figure 9.

------------------

(7) Section E on page 15-16 (including Algorithm 3, Table 7, and Figure 10), **Asynchronous Federated Continual Learning**. (**R1** and **R3**)
- We have performed additional experiments in an asynchronous federated continual learning setting, since different tasks may require different amount of time to train. Our FedWeIT can easy to extend for asynchronous scenarios using simple modifications. We have included the modified algorithm and experimental results for **asynchronous FedWeIT**, in Algorithm 3 and Table 7, in Section E of the Appendix.

---------------

We strongly believe that the problem we tackle (federated continual learning, inter-client interference between irrelevant tasks) and the idea (selective transfer of other task knowledge to each task) we propose to tackle the problem are both highly novel, and provide important contributions to the research in both continual learning and federated learning. We also believe that the new experimental results effectively address the concerns of the reviewers on the scalability of our method to a larger number of tasks, and its effectiveness under the asynchronous federated continual learning setting.

---

### Author Response · Authors · 2020-11-23
**Author Responses & Revision**

Dear Reviewers,

Could you please go over our responses and the revision since we can have interactions with you only by this Tuesday (24th)?
We have responded to your comments and faithfully reflected them in the revision, and provided additional experimental results that you have requested. We sincerely thank you for your time and efforts in reviewing our paper, and your insightful and constructive comments.

Thanks,
Authors

---

### Decision · Program_Chairs · 2021-01-07
**Final Decision**

**Decision:**

Reject

**Comment:**

This paper tackles an interesting problem (that of federated continual deep learning) and proposes an effective approach for it with good results.  This is a good contribution. However, there are presentation issues in several aspects of the paper that require improvement before publication. The authors' claims of the novelty of the federated continual learning problem is overstated (there was an AAMAS'18 paper they cited which IS applicable to the federated setting, although their method is certainly a substantial improvement, more flexible, and supports deep models), and there are aspects of the analysis and experiments that could be improved (the authors are somewhat dismissive of one reviewer's concerns about the insensitivity to the regularization parameters. While I agree with the authors that this aspect of the review is focusing on this one detail in lieu of the bigger picture, the author's insensitivity study in Figure 6 and subsequent analysis are lacking and could use improvement).  The authors' revisions did help clarify/address a number of issues raised in the initial reviews,  but it was felt that more improvement was needed.
In particular, there are still mistakes with characterizing this work with respect to existing work, especially in overstating the novelty of the federated continual learning problem.  I do believe this paper is the first to call it "federated continual learning" (a name I like), but the paper should be less dismissive of existing works on "multi-agent lifelong learning" that could also apply to this setting, albeit with shallow models.  Consequently, while this reduces the novelty of the federated continual learning problem, the authors still have a substantive contribution -- just one that needs improvement in presentation before publication.